# The Addition of Sterols and Cryoprotectants to Optimize a Diet Developed for *Eldana saccharina* Walker (Lepidoptera: Pyralidae) Using the Carcass Milling Technique

**DOI:** 10.3390/insects13040314

**Published:** 2022-03-23

**Authors:** Nomalizo C. Ngomane, John S. Terblanche, Des E. Conlong

**Affiliations:** 1Department of Conservation Ecology and Entomology, Stellenbosch University, Stellenbosch 7600, Western Cape, South Africa; nomalizo@riverbio.com (N.C.N.); jst@sun.ac.za (J.S.T.); 2South African Sugarcane Research Institute, 170 Flanders Drive, Mount Edgecombe 4300, KwaZulu-Natal, South Africa

**Keywords:** insect rearing, diet formulation, sterile insect technique, sugarcane, mass production, cholesterol, stigmasterol, cold tolerance, proline, trehalose

## Abstract

**Simple Summary:**

Despite being essential for growth, insects cannot synthesize sterols. Naturally, sterols are obtained from insects’ host plants, but in artificial diets are sometimes missing or not optimal, leading to reduced growth. By adding 1.0 g of stigmasterol, and a combination of 0.2 g each of cholesterol and stigmasterol to two standard minimum specification (MS) diets for *Eldana saccharina,* pupation was faster (72% and 70%, respectively) than the control diet (15%), 20 days after neonate inoculation. Nevertheless, a reproductive mass reared insect could become a poor field performer, as reproductive output and field performance might be traded off. To increase *E. saccharina*’s cold performance, cryoprotectants were added to a standard MS diet. Males from MS diets containing both concentrations of a proline/trehalose mix, and the highest concentration of trehalose, recovered 30–33% faster from chill coma than males from the remaining diets. Fertility of females that entered chill coma was reduced (<44%) when fed as larvae on cryoprotectant supplemented diets. Females not exposed to chilling treatment had 84% fertility when mated with males from the same source. The MS diet plus 0.2 g each of cholesterol and stigmasterol should become the standard diet, as larval development time was decreased by 60% without obvious trade-offs.

**Abstract:**

Various combinations and concentrations of cholesterol (C) and stigmasterol (S) were added into a base diet developed for *Eldana saccharina*. Survival of inoculated neonate was high on all diets (>92% at day 20 and >95% at day 27). Fastest larval development occurred on the minimum specification (MS) (+1.0 gS) and MS (+0.2 gC: 0.2 gS) diets (72 and 70% pupation respectively at day 20). Significantly slower development (15% pupation) occurred on the control diet at day 20. Female pupal weight increased when larvae fed on the MS (+0.1 gC), (+0.1 gS) and (+0.2 gC:0.2 gS) diets (0.2143 ± 0.00 g, 0.2271 ± 0.01 g and 0.2252 ± 0.01 g, respectively) as compared with the control diet (0.1886 ± 0.00 g). Adult emergence was significantly higher (100%) from the MS (+0.1 gS) and MS (+0.2 gC:0.2 gS) diets, as compared with the remaining sterol (95%) and control diets (97%). To potentially increase *E. saccharina*’s cold tolerance, inclusion of cryoprotectants L-proline (P) and trehalose (T) into the MS diet was investigated. Males from the MS (0.2 gP:0.2 gT), MS (0.5 gP:0.5 gT) and MS (1.0 gT) diets recovered fastest from chill coma treatment (204 ± 44 s, 215 ± 7 s and 215 ± 9 s, respectively) than those from the remaining cryoprotectant diets (305 ± 22 s). The addition of cryoprotectants severely reduced female fertility (<44%) when mated with non-chill coma exposed males. In contrast, eggs from females not exposed to chilling treatment were 84% fertile when mated with males from the same source. The MS (0.2 gC:0.2 gS) diet is the preferred choice to replace the currently used diet, reducing the larval growth period by 60% without negative effects on key life cycle parameters of *E. saccharina*.

## 1. Introduction

Rearing the African sugarcane stalk borer *Eldana saccharina* Walker (Lepidoptera: Pyralidae) on an artificial diet rather than on their natural host plants has proven beneficial for research purposes and pest control programs [1]. While rearing the pest under controlled conditions, it is important that the insects produced display characteristics similar in viability, vigour and behaviour to wild populations, and they should demonstrate corresponding resistance to pathogenic organisms [2]. The loss of vigour and viability of laboratory-reared populations is a well-known problem, often resulting in the collapse of insect colonies [2]. In many cases, after 4 to 5 generations, insect colonies experience reduced survival, growth rate, fecundity, fertility and hatching success, as well as difficulty in moulting under certain environmental conditions [2].

Sterols play an important role in the physiological processes of insects and greatly influence their growth and development [2,3]. They are essential components of plant and animal cell membranes and serve as hormone precursors and signalling molecules for phytophagous insects [2,4,5]. Cholesterol is the most common sterol in animals, while phytosterols are found in plants (e.g., sitosterol, stigmasterol) [6,7]. Unlike most animals, insects are unable to synthesize sterols de novo, and depend directly on a dietary supply of these nutrients [2,8]. Cholesterol incorporated into the diet usually satisfies the sterol requirements for most insect species [2,3]. Diets containing sitosterol and stigmasterol increased larval weight, growth rate and decreased mortality in diverse lepidopteran species [9]. A deficiency of sterols in artificial diets greatly contributes to reduced egg hatch, inability of insects to moult and increased mortality in early instars [2].

Furthermore, in insect rearing facilities, selection pressure is mostly directed towards parameters which are important for culture productivity e.g., high fecundity, high fertility, short life cycle, large size, etc., which is not necessarily a guarantee for optimal field performance, as needed for field releases in sterile insect technique (SIT) programmes. A high quality, reproductive insect reared in the laboratory could easily be a poor performer in the field, especially since reproductive output and field performance such as survival under stressful environmental conditions, flight ability and mating competitiveness might be traded off [10]. So, for most control methods in integrated pest management (IPM) approaches, some form of quality control or “filtering” for specific trait(s) is required to aid and/or enhance effective field performance or other required quality traits of laboratory-reared insects [11,12].

Temperature plays an important role in the development, activity, distribution, abundance and survival of insects in the field [11,13]. One crucial characteristic of climate identified to potentially limit species distribution is low-temperature performance or tolerance [14]. In tropical areas, insects exposed to 10–15 °C may result in a chill-coma or death, whereas insects in temperate and polar-regions may remain active and are able to fly at much lower, even sub-zero temperatures. Such insects compensate for potentially stressful temperatures by continuously adjusting their physiology and behaviour in order to survive and optimize their individual quality in the environment [15]. Depending on insect sensitivity, small changes in temperature may result in large differences in an insect’s metabolic rate or respiratory water loss, growth rate and phenology, partly since these effects are non-linear [14].

According to Kostal et al. [16], the process of cold acclimation and freezing tolerance of insects generally involves significant biochemical changes, such as rapid increases in concentrations of cryoprotectants (e.g., amino acids and carbohydrates) and increases in relative proportions of phospholipids that couple palmitic and linoleic fatty acids in cell membranes. The rapid increases in these likely contribute to the preservation of proteins and membrane structures, and insect function at low temperatures [16].

Artificial diets for mass-rearing *E. saccharina* have been developed [17,18,19,20,21,22,23,24] and these diets effectively supported optimum survival and development of *E. saccharina* for the purposes for which they were developed. Growth parameters and biological quality traits of the pest have also been investigated by the authors on their developed diets. However, little is known about how the current and past laboratory diets, or host plants, influenced low temperature tolerance of *E. saccharina*. This understanding is important, as it influences field and low temperature activity, as well as mass-rearing and subsequent field performance. It could even explain the importance of being cold temperature tolerant while living in a sub-tropical climate.

To further this knowledge, this study investigated two aspects: First, whether sterols (i.e., cholesterol and stigmasterol) added to a minimum specifications (MS) diet, developed using the carcass milling technique (CMT) [23,24] had an added beneficial effect on the growth, development and reproduction of *E. saccharina*, with the aim to further improve its mass-rearing; and second, whether cold hardiness of *E. saccharina* moths could be enhanced through the addition of selected cryoprotectants (i.e., proline and trehalose) into the MS diet.

## 2. Materials and Methods

### 2.1. Experimental Site

The study was undertaken at the South African Sugarcane Research Institute (SASRI) Insect Rearing Unit (IRU), Mount Edgecombe, KwaZulu-Natal, South Africa. Laboratory standard operational procedures developed for the SASRI-IRU were followed to prevent contamination and contamination spread in the diets being tested [22]. All temperature and relative humidity (RH) conditions were kept similar for all aspects tested for both the formulated and control diets (26 ± 2 °C, 72 ± 5% RH). Photoperiod was different for rearing (0 Light (L): 24-h Dark (D)), quality (8-h L: 16-h D), chill coma recovery (conducted during the day) and oviposition assessments (8-h L: 16-h D). The larvae were exposed to 0 L: 24-h D photoperiod because they are the life stage being subjected to the diets. Generally, they feed cryptically inside stalks where no light reaches them. Chill coma recovery and oviposition trials, on the other hand, were conducted on adults during the 8-h L: 16-h D photoperiod, because adults are exposed to daylight in their natural life cycle.

### 2.2. Eldana saccharina Rearing

#### 2.2.1. Formulation for Sterol Incorporation into Diets

Table 1 describes the MS diet recipes containing different concentrations of cholesterol (C) and stigmasterol (S) (MS (0.1 gC), MS (1.0 gC), MS (0.1 gS), MS (1.0 gS), MS (0.2 gC:0.2 gS) and MS (0.5 gC:0.5 gS)). The cholesterol additive was cholest-5-en-3β-ol (cholesterol, ≥99%) and the stigmasterol additive was stigmasta-5,22e-dien-3β-ol (stigmasterol, ≥98%), purchased from Sigma-Aldrich Chemicals Company, Missouri, United States. The control was a modified European corn borer *Ostrinia nubilalis* Hubner (Lepidoptera: Pyralidae) diet (ECBMOD) currently used to routinely rear *E. saccharina* [22]. The MS diets were formulated according to minimum nutrient specifications for mass-rearing *E. saccharina* as determined by Woods et al. [23].

#### 2.2.2. Formulation for Cryoprotectant Incorporation into Diets

The minimum specification (MS) diet was used again as the base diet into which two different synthetic cryoprotectants, the amino acid proline (L-proline ((s)-pyrrolidine-2-carboxylic acid (P) Product number PO380, ReagentPlus^®^, ≥99% (HPLC)) and carbohydrate trehalose (α-d-glucopyranosyl-α-d-glucopyranoside (T) Product Number T9531, from brewers’ yeast *Saccharomyces cerevisiae,* ≥99%) (Both purchased from Sigma-Aldrich Chemicals Company, St. Louis, Missouri, USA), at different concentrations were added (MS (0.1 gP), MS (1.0 gP), MS (0.1 gT), MS (1.0 gT), MS (0.2 gP:0.2 gT), MS (0.5 gP:0.5 gT)). The modified *O. nubilalis* diet (ECBMOD) was used as the control diet. Their formulations are given in Table 2.

#### 2.2.3. Diet Preparation

According to the formulated diet recipes listed in Table 1 and Table 2, ingredients were weighed using a calibrated Mettler Toledo New Classic ML6001 balance with accuracy of 0.01 mg (Microsep (Pty) Ltd., Johannesburg, South Africa). The weighed dry content ingredients of each formulated diet was poured into the bowl of a 6.7 L Kenwood Titanium Major KMM060 food mixer and thoroughly mixed for one minute. A ratio of 1:3 boiling water (500 g:1500 mL dry matter to water), together with 8.00 mL of acetic acid, was added to the mixture. The food mixer was further allowed to run for one minute. The resulting mixture was poured into a 2.5 L plastic mixing bowl and placed in a 700 W AMW17 manual microwave oven to cook for 2 min on high heat. Extra care was taken to ensure that the diet did not overcook. As soon as the diet started bubbling within the 2 min, it was removed from the microwave, thoroughly mixed using a sterilised spoon (dipped in Denol (70%) and then in distilled water) and then placed back in the microwave to continue cooking until the 2 min-cooking period was completed.

A similar procedure as described above was used to prepare the control diet, with the exception that the agar solution was prepared separately. Five hundred ml of boiling water was dispensed in a 1 L plastic jug, into which 4.6 g of agar powder was poured slowly and stirred using a spoon to avoid the formation of agar lumps. The agar solution was poured into a 1 L Schott Pyrex bottle and autoclaved for one hour at 121 °C. Once autoclaved, the hot agar was poured into the running food mixer, and a balance of 1000 mL of boiling water with 8.00 mL of acetic acid was added to the autoclaved agar solution, followed by the weighed dry mixture. The resulting mixture was poured into a 2.5 L plastic mixing bowl and placed in a 700W AMW17 Manual microwave oven on high heat to cook for 2 min, as described before.

#### 2.2.4. Diet Dispensing

For each formulation described in Table 1 and Table 2, seven hundred 25 mL plastic screw top vials (PerkinElmer, Thomas Scientific, Pretoria, South Africa) were sterilised overnight in a 0.5% sodium hypochlorite (NaOCl) solution. They were shaken to remove most of the NaOCl solution before being placed on a running laminar flow bench (Constructed locally at the Mechanical Engineering workshop of SASRI) to dry. They were further surface sterilized with ultraviolet germicidal lights behind ultraviolet (UV) resistant welding curtains on the laminar flow bench. After being cooked, but before the diets could set, 10 mL of diet was dispensed into each vial using a Jencons Scientific Perimatic GP II Peristaltic Pump Dispenser (Cambridge Scientific Products, Watertown, MA, USA). A total of 100 vials were replicated for each diet formulation. Fifteen vials prepared per diet formulation were used to determine development time to first pupation and the remaining 85 vials were used for harvest at full pupation. The vials with their dispensed diet were left to cool on a running laminar flow bench for an hour. Once cool, the diet surface in each vial was scarified with a sterilised dissecting needle (dipped in Denol (70%) and then in distilled water), allowing the enclosed larvae to access the diet more easily. In each vial, two grams of sago (starch extracted from pith of various tropical palm stems and formulated into dry pearl like balls, commonly sold as a commercial product in local supermarkets), mixed with 0.0004 g of Dithane M45 as a fungicide, was poured over the surface of the diet. The sago helped absorb excess moisture on the surface of the diet and also served as a refuge for neonate larvae before they entered the diet.

#### 2.2.5. Inoculation of Neonate Larvae onto the Diet

*Eldana saccharina* eggs oviposited on sheets of single ply paper towelling (270 × 1500 mm, Wisper, Epping Industrial Supplies, Cape Town, South Africa) were collected from the SASRI-IRU adult emergence and oviposition room. The eggs were kept at 24 °C, 72% RH and 0 L: 24-h D photoperiod in a CLN 32 Laboratory Incubator (Pol-Eko Aparatura, Wodzislaw Slaski, Poland) for 7 days, for them to hatch into neonate larvae. In each vial, two neonate larvae were carefully placed on top of the sago, using a fine paintbrush (Winsor & Newton Cotman 111 round, size 0000) dipped in Denol and then distilled water. To prevent neonates from escaping, the vials were sealed with lids that provided proper ventilation through stainless steel fine mesh gauze. They were labelled (i.e., diet formulation, inoculation date, quality assessment dates), placed in plastic storage baskets and kept in a larval growth room at 26 ± 2 °C, 72 ± 5% RH and 0 L: 24-h D photo phase. The plastic storage baskets were stacked on clean 5-tier metal racks, and a Maxim iButton DS1923 (programmed using the Fairbridge Technologies ColdChain thermodynamics software) (Fairbridge Technologies, Sandton, South Africa) was placed in the larval growth room with the inoculated vials, hanging on a tag on a 5-tier metal rack, to monitor the temperature and humidity in the room at 15-min intervals.

### 2.3. Quality Assessment

#### 2.3.1. Development Time to First Pupation

After 20 days in the larval growth room, *E. saccharina* life stages from the allocated 15 vials were manually and gently extracted from the diets in the vials using dissecting forceps and divided into size and life cycle categories (1st/2nd instar, 3rd/4th instar, 5th/6th instar, pre-pupal stage and pupal stage). These were placed into one of five 250 mL plastic jars with lids, according to its respective label. Each of the test diets were assessed in this way. The respective life stages were counted and recorded after the diet assessment was completed. Dead insects from each diet were counted and recorded as they were found.

#### 2.3.2. Pupal Harvesting and Pupal Weighing

The remaining 85 vials per formulation were harvested 27 days after inoculation. Details of the harvested vials were recorded (i.e., diet formulation, date of inoculation, date of harvest and number of vials harvested) and the insect life stages (i.e., larvae, pre-pupa, pupae, moths and dead insects) collected were counted and recorded. To prevent adult escape, harvested pupae were placed individually into cells of multicell trays (*n* = 32 × 20 mL cells per tray) covered with ventilated plastic cling wrap film (Thomas Scientific, Pretoria, South Africa). Pupae were placed singly in each cell to ensure that virgin males and virgin females were available for further experiments (i.e., oviposition testing). Trays were labelled with diet formulation and harvested date, stacked on multicell tray metal racks and stored in the IRU adult emergence and oviposition room, at 26 ± 2 °C, 72 ± 5% RH and 8:16 L:D photoperiod. The trays were stacked on 5-tier metal racks to ensure sufficient ventilation and even temperature and humidity distribution.

For each diet formulation, pupae were carefully cut out of their cocoons collected from harvested batches of pupae. Using a dissecting microscope (Zeiss, Stemi 2000-c, X600), 30 male and 30 female pupae per formulation were identified, based on the different structures of their external genitalia [25]. The pupae of both sexes were separately weighed using a calibrated balance with an accuracy of 0.0001 mg (Mettler Toledo ML54 Analytical Balance) (Microsep (Pty) Ltd., Johannesburg, South Africa).

#### 2.3.3. Adult Emergence and Sex Ratio

Pupae produced from the different diets were used to assess adult emergence and sex ratio. Freshly emerged adult males and females were counted and recorded the day after harvest and on a daily basis thereafter (process took a maximum of 2 weeks). To identify and separate insects that had emerged, markings on the cells (M for males and F for females) were made. From the emergence, the ratio of males-to-females in each diet formulation was determined.

#### 2.3.4. Chill Coma Recovery Time

Chill coma recovery was conducted to determine cold hardiness on *E. saccharina* moths reared on the different diet formulations. Freshly emerged male and female moths from each diet formulation were placed individually into watertight plastic vials (35 mL) (PerkinElmer, Thomas Scientific, Pretoria, South Africa). A total of 30 males and 30 females per diet formulation were used. Each individual exposed to chill coma was considered as a repetition. The vials were sealed with solid, unperforated lids and labelled accordingly (i.e., diet formulation and moth sex). The vials with moths were submerged in an ice-water slurry in a polystyrene cooler box for 2 h. Temperature of the ice-water slurry (i.e., 0 °C) was confirmed with a NIST-certified thermometer. At the 2-h mark, the vials were removed from the ice-water slurry. The moths were individually removed from their vials and placed on their backs in 9 cm diameter petri dish bases on a laboratory bench at 26 °C and 72% RH. The time taken to recover (i.e., regain muscle control, turn over and stand upright on their legs again) from the cold stress was scored to the nearest second as chill coma recovery time [26].

### 2.4. Oviposition Testing

#### 2.4.1. Male and Female Mating Frequency

To determine mating frequency of the males emerging from each diet, a freshly emerged virgin male and female moth pair were placed into a 500 mL paper drinking cup containing a pleated cardboard oviposition substrate (50 × 10 mm when pleated 5 times) held together with a paper clip. The paper cup lids were fixed with a 10 mm dental cotton wick soaked in distilled water for adults to drink from [21]. After the moth pair and oviposition substrate was placed in the paper cups, the lids were placed on top and labelled accordingly (i.e., diet formulation, harvested date, inoculated date). The paper cups were placed upright in plastic storage baskets and placed on 5-tier metal racks in the IRU adult emergence and oviposition room [21].

The oviposition substrate and female moth were removed on a daily basis, and the water supply was replenished. A new oviposition substrate and a freshly emerged female was placed into the paper cup with the remaining male and left to mate overnight [21]. In this section, the oviposition substrate was not used for fecundity but was placed in the cup to prevent the females from randomly laying eggs inside the cups. The removed female was killed by freezing and placed into a plastic Ziplock resealable bag with the male she was paired with and the date she was placed with the male. The female was then dissected under a dissecting microscope to determine her mating status by checking for spermatophores in her *bursa copulatrix* [21]. The procedure was repeated with all the females presented to each male until the male died, to determine how many females the males from each diet successfully copulated with in its lifetime. A total of fifteen males per diet formulation were used. A similar procedure was used for freshly emerged virgin males who had been subjected to chill coma treatment, mated with virgin females who had not been subjected to the chilling treatment.

To determine female mating frequency, virgin female moths were paired with virgin male moths from each diet formulation (prepared as described above). Daily, the oviposition substrate and the male moth was removed, and the water supply was replenished. A new oviposition substrate and freshly emerged male moth was placed into the paper cup with the remaining female moth and left to mate overnight [21]. This procedure was repeated until the female died, after which she was dissected to assess mating frequency by counting the spermatophores in her bursa copulatrix, assuming the males she was paired with transferred only one spermatophore to her *bursa copulatrix* on the night with her [21]. A total of 15 females per diet formulation were used to assess mean mating frequency. A similar procedure was used for freshly emerged virgin females who had been subjected to chill coma treatment, mated with virgin males who had not been subjected to the chilling treatment.

#### 2.4.2. Female Fecundity and Fertility

To determine female fecundity, a single freshly emerged male and female moth pair was placed into a 500 mL paper drinking cup, prepared as described in the mating frequency section above. A total of 15 moth pairs per diet formulation were used. On a daily basis, oviposition substrates were replaced and the water supply was replenished until the female moth died [21]. Each collected oviposition substrate was inserted into a plastic resealable Ziplock bag, labelled accordingly (i.e., diet formulation, inoculated date, cup number, date of removal) and placed in an incubator at 24 °C, 72% RH and zero L: 24-h D photoperiod. To determine mean fecundity per *E. saccharina* female, total number of eggs laid on the oviposition substrates each day were counted under a dissecting microscope for each female in the trial. These were summarized for each female and divided by the number of females in the trial to get the mean fecundity of each female in the trial [21].

To determine mean fertility of the eggs oviposited per female, total number of black head stage eggs [27] or neonate larvae emerging from the eggs oviposited by each female each day, were counted under a dissecting microscope, 5 days after each oviposition substrate was removed from the cup, to allow the neonate larvae to develop in the eggs. Unfertilised eggs were also counted. The fertile and unfertile eggs oviposited per day per female were summed, and percentage fertilised eggs were calculated per female. These were then summarized for each female and divided by the number of females in the trial to get the mean fertility of the eggs oviposited by each fertilised female in the trial [21]. A similar procedure for determining fecundity and fertility was used for females who had been subjected to chill coma treatment and had mated with virgin males who had not been subjected to the chilling treatment.

### 2.5. Statistical Analysis

One-way analysis of variance (ANOVA) was performed using IBM SPSS Statistics version 22 (2013: IBM Corp., Costa Mesa, CA, USA) on percentage survival and population age distribution (at the time to first pupation and at full pupal harvest), male and female pupal weight, moth emergence and sex ratio, male and female chill coma recovery time, and female fecundity and fertility data to compare means between the different diet formulations listed in Table 1 and Table 2. Significant means were separated with a Tukey’s HSD post hoc test, at *p* < 0.05. Key assumptions of ANOVA were checked and met for homogeneity of variance (using the Levene’s test, at *p* > 0.05) and normality (using a Shapiro-Wilk test for normality) of data distributions. A two-sample *t*-test (*p* < 0.05) was performed, using IBM SPSS Statistics version 22, on male and female mating frequency and fecundity and fertility data from the diets listed in Table 2 to compare means between chill coma exposed moths and those that were not exposed to the chilling treatment. To test whether the data sets for chill coma exposed and non-exposed moths were normally distributed, a W-test for normality was used. *p* values less than 0.05 were deemed to be significantly different.

## 3. Results

### 3.1. Insect Production during Quality Control and at Harvest

#### Diet Incorporation of Sterols

There were no significant differences observed in survival of neonates inoculated on all diets. Overall, survival was good, with a mean of 98 ± 1% [*n* = 15] (*p* = 0.108) of the neonates developing through to 5th and 6th instar larvae, pre-pupae and pupae in the time to first pupation trial, and a mean of 99 ± 0% [*n* = 85] (*p* = 0.065) of these life stages surviving up till full pupal harvest (Table 3).

### 3.2. Population Age Distribution

#### Diet Incorporation of Sterols

The incorporation of sterols into the MS diet, showed significant effects on *E. saccharina* larval development, irrespective of the type and concentration of sterol added, as compared with the control diet (ECBMOD). On day 20 after inoculation, all neonates inoculated on the MS diets with sterols, showed fast larval development. This is evident by the higher percentage of pupae (≥55%), lower percentage of the 5th/6th instar larvae (≤33%) and no smaller instar (1st/2nd and 3rd/4th instar) larvae, whereas larval development in the ECBMOD diet was much slower producing a higher percentage of the 3rd/4th instar (35%), 5th/6th instar larvae (50%) and fewer pupae (15%) (Table 4).

The cholesterol containing MS (1.0 gC) diet produced the highest proportion of pupae (96%) at harvest after 27 days, and if % adult emergence (4%) was added to the % pupation, 100% of the neonates inoculated had pupated and had some adults emerge from the pupae. In this diet, there were no longer any larvae found. In contrast, only 70% of the neonates inoculated had pupated on day 27 in the ECBMOD control diet. At this time, 19% of the life stages in the ECBMOD control diet were still larvae, as compared with between 0 and 7% in the sterol containing MS diets. The MS diet containing the higher concentration of the sterol mix (0.5 gC:0.5 gS) had a similar pupation rate (73%) to the control, and the lowest pupation for all the sterol containing diets (Table 5).

### 3.3. Male and Female Pupal Weight

#### 3.3.1. Diet Incorporation of Sterols

In all diet formulations, female pupae were significantly heavier than male pupae. The addition of different sterols and concentrations of sterols did not result in any significant differences in male pupal weights between diet formulations (0.12 ± 0.00 g [*n* = 30]; *p* = 0.062). However, there were no significant differences in female pupal weights derived from the experimental diets MS (0.1 gC) (0.21 ± 0.01 g [*n* = 30]; *p* = 0.042), MS (0.1 gS) (0.23 ± 0.01 g [*n* = 30]; *p* < 0.001) and MS (0.2 gC:0.2 gS) (0.23 ± 0.01 g [*n* = 30]; *p* = 0.001), but these pupae were significantly heavier than those from the ECBMOD control diet (Figure 1).

#### 3.3.2. Diet Incorporation of Cryoprotectants

There were no significant differences observed in male pupal weights produced from the diet formulations (weight: 0.13 ± 0.00 g [*n* = 30]; *p* = 0.054), and these were significantly lighter than female pupae. However, there were significant differences in female pupal weights between diet formulations. The weight of female pupae produced from the MS (0.1 gP) (weight: 0.23 ± 0.01 g [*n* = 30]; *p* = 0.002) and MS (0.1 gT) (weight: 0.22 ± 0.01 g [*n* = 30]; *p* = 0.009) diets were not significantly different, but they were significantly heavier than those from the MS (1.0 gT) diet (Figure 2).

#### 3.3.3. Evaluation of *Eldana saccharina* Performance between Diets

For all the diets tested in the trials, female pupal weight was close to double that of male pupal weight (Figure 1 and Figure 2). When sterols were incorporated into the MS diet developed using the CMT, female pupae were heavier than those produced in the control diet (Figure 1). Female pupae from the sterol diets were also heavier than those from the cryoprotectant diets, apart from the MS (0.1 gP) and MS (0.1 gS) diets, which also produced heavier females (Figure 2). Male pupal weights from the cryoprotectant diets were heavier than those produced from sterol diets, apart from the MS (0.1 gC) and MS (0.5 gC:0.5 gS), which also produced heavier males (Figure 1).

### 3.4. Moth Emergence and Sex Ratio

#### 3.4.1. Diet Incorporation of Sterols

Moth emergence from pupae harvested from all diet formulations was high (≥92%). The highest emergence (100 ± 0.3% [*n* = 170]; *p* < 0.034) was from pupae harvested from the MS (0.1 gS) and MS (0.2 gC:0.2 gS) diet formulations, followed by those from pupae from the control (97 ± 0.6% [*n* = 170]; *p* < 0.034), MS (0.1 gC) (96 ± 0.6% [*n* = 170]; *p* < 0.003) and MS (1.0 gC) (96 ± 0.6% [*n* = 170]; *p* < 0.003) diets. The sex ratio of adults emerging from pupae harvested from the ECBMOD control (0.8 ± 0.1 [*n* = 170]; *p* < 0.031), MS (0.1 gS) (0.8 ± 0.1 [*n* = 170]; *p* < 0.031) and MS (1.0 gS) (0.9 ± 0.1 [*n* = 170]; *p* < 0.031) diets favoured more females than males as compared with pupae from the remaining diets (Table 6).

#### 3.4.2. Diet Incorporation of Cryoprotectants

Moth emergence from pupae from the different cryoprotectant diet formulations was again high (≥89%), with pupae harvested from the MS (1.0gT) (100 ± 0.3% [*n* = 170]; *p* < 0.005) all emerging as adults, followed by pupae harvested from the MS (1.0gP) (96 ± 0.6% [*n* = 170]; *p* < 0.005) and MS (0.5gP:0.5gT) diets (96 ± 0.6% [*n* = 170]; *p* < 0.005) and the MS (0.1gP) (95 ± 0.6% [*n* = 170]; *p* < 0.002) diet. Apart from the sex ratio of *E. saccharina* moths obtained from pupae harvested from the ECBMOD control diet, which produced more males than females (1.7 ± 0.1 [*n* = 170]; *p* < 0.003) and the MS (0.1gP) diet, which produced more females than males (0.9 ± 0.1 [*n* = 170]; *p* < 0.003), the sex ratio of adults emerging from the remaining diet formulations, were all close to 1, meaning that the sex ratios are not biased towards males nor females (Table 7).

#### 3.4.3. Evaluation of *Eldana saccharina* Performance between Diets

Moth emergence from the harvested pupae in all diets tested in the trials (Table 6 and Table 7) was high (≥89%) and the sex ratio of moths emerging from pupae was close to 1, except for the sex ratio of the control diet produced in the cryoprotectant incorporation trial, where more males were produced than females and the MS (0.1 gP) diet, which produced more females than males. It is thus apparent that the addition of sterols or cryoprotectants to the MS diet formulation determined from the CMT had no effect on the emergence of adults from the pupae harvested from all the diet formulations, nor on their sex ratio.

### 3.5. Chill Coma Recovery Time

#### Diet Incorporation of Cryoprotectants

Females from all diet formulations had no significant differences in chill coma recovery time, all being around 291 ± 37 s [*n* = 30] (*p* = 0.237). However, there were significant differences in male chill coma recovery time. Males from the MS (0.2 gP:0.2 gT), MS (0.5 gP:0.5 gT) and MS (1.0 gT) diets recovered faster from the exposed chill coma (204 ± 44 s [*n* = 30]; *p* ≤ 0.041, 215 ± 35 s [*n* = 30]; *p* ≤ 0.018 and 215 ± 7 s [*n* = 30]; *p* ≤ 0.041, respectively) than males from the MS (1.0 gP), MS (0.1 gT) and MS (0.1 gP) (295 ± 9 s [*n* = 30]; *p* = 0.041, 307 ± 28 s [*n* = 30]; *p* ≤ 0.041 and 314 ± 29 s [*n* = 30]; *p* ≤ 0.022, respectively) (Figure 3).

### 3.6. Male and Female Mating Frequency

#### Diet Incorporation of Cryoprotectants

Comparisons of mating frequency between moths exposed to chill coma and those that were not exposed to the chilling treatment, from each diet formulation, were made. There were no significant differences observed in mating frequency of males exposed to chill coma and those not exposed to chill coma from each diet formulation (MS (0.1 gP): t (28) = −0.277, *p* = 0.795; MS (1.0 gP): t (28) = −1.414, *p* = 0.230; MS (0.1 gT): t (28) = −1.342, *p* = 0.251; MS (1.0 gT): t (28) = −1.000, *p* = 0.374; MS (0.2 gP:0.2 gT): t (28) = −0.447, *p* = 0.678; MS (0.5 gP:0.5 gT): t (28) = 0.500, *p* = 0.643; ECBMOD: t (28) = −2.121, *p* = 0.101) (Table 8).

There were no significant differences observed in mating frequency of females exposed to chill coma and those not exposed to chill coma from each diet formulation (MS (0.1 gP): t (28) = 1.000, *p* = 0.374; MS (1.0 gP): t (28) = 0.000, *p* = 1.000; MS (0.1 gT): t (28) = 0.707, *p* = 0.579; MS (1.0 gT): t (28) = −1.414, *p* = 0.230; MS (0.2 gP:0.2 gT): t (28) = −0.894, *p* = 0.422; MS (0.5 gP:0.5 gT): t (28) = −0.894, *p* = 0.422; ECBMOD: t (28) = 0.001, *p* = 1.000) (Table 8).

### 3.7. Female Fecundity and Fertility

#### 3.7.1. Diet Incorporation of Sterols

There were no significant differences observed in fecundity (878.00 ± 24.44 eggs [*n* = 15]; *p* = 0.314) and fertility (96.00 ± 1.07% [*n* = 15]; *p* = 0.599) of female moths produced from the different diet formulations when mated with male moths from the same diet formulations (Table 9).

#### 3.7.2. Diet Incorporation of Cryoprotectants

Comparisons of fecundity and fertility between females exposed to chill coma and those that were not exposed to the chill coma and mated with non-chill coma-exposed males from each diet formulation, were made. There were no significant differences observed in fecundity of females exposed to chill coma and those not exposed to chill coma from each diet formulation (MS (0.1 gP): t (28) = −1.701, *p* = 0.224; MS (1.0 gP): t (28) = 0.659, *p* = 0.553; MS (0.1 gT): t (28) = −0.180, *p* = 0.870; MS (1.0 gT): t (28) = −0.034, *p* = 0.976; MS (0.2 gP:0.2 gT): t (28) = −0.289, *p* = 0.793; MS (0.5 gP:0.5 gT): t (28) = −0.052, *p* = 0.963; ECBMOD: t (28) = −0.323, *p* = 0.767) (Table 10). However, there were significant differences observed in fertility. Fertility of females, produced from the MS (0.1 gP), MS (1.0 gP), MS (0.2 gP:0.2 gT) and ECBMOD diets that were exposed to chill coma, was significantly reduced when compared to the fertility of females from the same formulations that were not exposed to the chilling treatment (MS (0.1 gP): t (28) = −12.562, *p* < 0.001; MS (1.0 gP): t (28) = −9.105, *p* = 0.001; MS (0.2 gP:0.2 gT): t (28) = −4.620, *p* = 0.033; ECBMOD: t (28) = −5.014, *p* = 0.009) (Table 10). It is evident that females exposed to chill coma, once recovered and mated with non-chill coma exposed males, irrespective of cryoprotectant treatment, had the fertility of their eggs severely compromised as compared with the fertility of females not exposed to chill coma.

#### 3.7.3. Evaluation of *Eldana saccharina* Performance between Diets

The addition of sterols into the MS diet formulation did not seem to have any impact on fecundity and fertility of female moths produced from the different formulations. The females produced from these diets oviposited a similar number of eggs (>870 eggs) that were more than 90% fertile (Table 9), to that of females produced on the control diet. Fecundity of females produced from the diets containing cryoprotectants (Table 10), irrespective of the females being subjected to cold treatment or not, was similar to that of females produced by the control diet and sterol containing diets. Although fecundity of females produced from the cryoprotectant diets was not affected by the chilling treatment, fertility of chill coma-exposed females, irrespective of the diet formulation they came from, was severely compromised, even though they were mated with males from the same source as the non-chill exposed females.

## 4. Discussion

### 4.1. Improved Eldana saccharina Growth, Development and Reproduction through the Incorporation of Sterols into the CMT Derived MS Diet Formulation

The inclusion of sterols, which are essential to insect growth and development [3] and have improved insect performance when fed to them in artificial diets [2,8,9], was investigated with the aim to further improve the minimum specifications (MS) artificial diet for *E. saccharina*. Based on studies with other insects [2,9], the type and concentration of sterols incorporated into the MS diet in this study were deemed to be optimal with respect to survival, growth and development.

#### 4.1.1. Insect Development

Survival of *E. saccharina* larvae reared on sterol enriched diets did not differ significantly from that of *E. saccharina* reared on the control diet (EDBMOD). However, the larval period was significantly shortened in the sterol diets, irrespective of the type and concentration of sterol added. This reduction in larval development period of *E. saccharina* reared on sterol enriched diets suggests that the current routine and MS diets are deficient in key insect sterols. These should be incorporated into the MS diet for better and faster development of *E. saccharina*, thereby effectively improving production of the insect and reducing costs of mass-rearing without impacting on other life cycle traits.

More specifically, Al-Izzi and Hopkins [9] found that female larvae developed to pupae more rapidly as concentrations of sitosterol and stigmasterol increased, but increased cholesterol levels did not affect the larval period, with development not as rapid as with the phytosterols. They also observed that at lower concentrations of sitosterol and stigmasterol, female larvae developed at a significantly slower rate than did males. In the current study, the sexes were not separated; however, similar observations to that of Al-Izzi and Hopkins [9] were made. At a higher concentration of stigmasterol (1.0 g) *E. saccharina* larval period was significantly reduced as compared with larvae from the diets containing cholesterol. The current study also demonstrated that the sterol mix at the lowest concentration (0.2 gC:0.2 gS) in the diet reduced *E. saccharina*’s larval period, and this was similarly demonstrated by Babu et al. [2] who observed pupation rates and emergence to be better on diets containing all sterols (i.e., cholesterol, stigmasterol and sitosterol).

#### 4.1.2. Pupal Weight

A further indication that the addition of sterols to the routine diet would benefit the rearing of *E. saccharina* was that pupal weights of females fed as larvae on the diets incorporated with lower concentrations of cholesterol and stigmasterol (0.1 g) and a combination of the lower mix of sterols (0.2 gC:0.2 gS) were significantly increased, as compared with the control diet. This has important implications for the mass-rearing and thus an SIT program, in that fewer females will be needed because of their increased size and thus fecundity [21], to produce high numbers of good quality insects [2].

The incorporation of sterols into the MS diet, irrespective of the type and concentration of sterol added, did not impact *E. saccharina* male pupal weight, suggesting that males may have a lower sterol requirement than females [9].

### 4.2. Enhanced Cold Hardiness of Eldana Saccharina Moths Due to the Incorporation of Cryoprotectants into the CMT Derived MS Diet Formulation

Low temperature has been identified to potentially limit insect species distribution [14]. All insects have a preferred range of temperature at which they will thrive, and at lower temperatures, they become less active until they eventually lose all mobility [28]. A gradual decline in temperature, however, coupled with microclimates caused by daily or seasonal temperature fluctuations, might prepare the insect to tolerate freezing temperatures [28].

In the case of laboratory-reared insects, their field performance may be compromised by their rearing conditions. This may occur due to laboratory adaptation, inbreeding depression, unintended selection or through direct rearing effects (e.g., crowding and artificial diet) [12,29]. Therefore, it is important to establish quality control or filtering for specific trait(s) when rearing insects for field releases, to aid and/or enhance effective field performance or competitiveness of these laboratory-reared insects [11,12].

One way that laboratory-reared insects can survive cold conditions not commonly encountered in their rearing environment, is through the accumulation of cryoprotectants, such as sugars and amino acids, that increase their cold tolerance by lowering the freezing point of liquids in the haemolymph [16]. To date, a handful of studies have focused on the ability to artificially improve the cold tolerance of an insect that is freeze intolerant and very few studies have attempted to improve the cold tolerance of insects destined for release in an SIT programme [16,30,31]. In addition to the necessity to improve the cold tolerance of insects for the SIT, it is important to consider how cryoprotectants may affect other aspects of the insect’s physiology.

In the current study, cryoprotectants were incorporated into the MS diet to investigate whether they improved cold hardiness of *E. saccharina* moths that fed on the diet as larvae, for effective field releases. Based on studies with other insects [30], the types and concentrations of cryoprotectants selected for this study were found to be optimal in terms of egg production and viability, flight ability and cold tolerance.

#### 4.2.1. Chill Coma Recovery Time

The natural geographic distribution of *E. saccharina* in South Africa stretches across a wide range of climates, but predominantly in the subtropical climates found in KwaZulu-Natal and sub-Saharan Africa [14,32], where temperatures do not fluctuate widely. In temperate regions in contrast, temperature fluctuates markedly on a daily and seasonal basis, requiring insects from these regions to be pre-adapted to frequent and unpredictable thermal variations. They should also be able to recover quickly from chill injuries that may occur during unpredictable cold periods [33]. So far, considerable attention has been given to the capacity of certain insects to survive extreme freezing temperatures, and the induction of antifreeze compounds, or cryoprotectants, that have been identified in many species [34]. However, cryoprotection is not the only cold tolerance mechanism, as many species die at temperatures above their freezing point. In this regard, studies have analyzed the survival of insects at stressing temperatures far above their freezing point [34,35].

When insects are exposed to low temperatures, provided that cold treatment is not too long, they enter a reversible cold-induced paralysis driven by a failure of neuromuscular function [36]. In this state, insects cannot pursue activities critical for their fitness such as feeding, mating or oviposition. Chill coma onset has been associated with the loss of ionic and osmotic homoeostasis during low-temperature exposure [36]. Recovery from chill coma requires a re-establishment of metabolic homoeostasis, recovery of ionic and water balance and repair of critical cellular components [36]. Insect diet components that include high carbohydrate and protein content are known to help restore metabolic homeostasis and energy balance of cold exposed insects that have fed on these diets, thus increasing their cold tolerance [30,37].

The results in this study demonstrated that the incorporation of cryoprotectants into the MS diet enhanced *E. saccharina* adult male cold tolerance in that it reduced the chill coma recovery time (i.e., recovered faster). The diet that gave the best results, outperforming the control diet, was the MS diet containing the lower concentration of the cryoprotectant mix (0.2 gP:0.2 gT), followed by the diets containing a higher concentration of trehalose (1.0 gT) and higher concentration of the cryoprotectant mix (0.5 gP: 0.5 gT). Rearing *E. saccharina* on these formulations has important implications for an SIT program in that sterile males will be able to recover faster than wild males from chilling temperatures if encountered in release sites. The chilling treatment did not affect mating frequency of *E. saccharina* males and females reared on cryoprotectant supplemented diets. This again has important implications for an SIT program in that the released males exposed to chilling temperatures will be able to mate successfully with wild females after they have recovered from chill coma exposure [38].

The *E. saccharina* mass-rearing program can benefit from adopting a low temperature treatment as part of their operating procedure since the program requires storage of sterile insects in cold temperature without loss of performance and to mobilize them quickly upon demand [33]. Sterile insects are also exposed to low temperature during shipping from mass-rearing facilities to release sites [38]. Lastly, temperatures within release sites are different from rearing conditions and may be stressful to released insects, thus implementing thermal conditioning protocols before release may be useful in improving the survival of insects during low temperature exposures [33,38].

#### 4.2.2. Fecundity and Fertility

The cryoprotectant diets did not have any pronounced effect on fecundity of *E. saccharina* females, irrespective of being subjected to cold treatment or not, and fertility of females not exposed to the chilling treatment. The fecundity and fertility of females not exposed to chill coma treatment was found to be similar to that of the control and sterol containing diets, proving that these diets are of similar quality and are suitable for rearing *E. saccharina*. However, fertility of chill coma exposed females seemed to be severely compromised, resulting in reduced fertility (≤43%) when compared with the control and sterol diets, when mated with males not exposed to chill coma conditions. According to Findsen et al. [36] and Mensch et al. [39], long-term exposure to low temperatures during adult maturation might decrease fertility after cold recovery as a result of carry-over effects on reproductive tissues. Exposure at 0 °C for 2 h may have caused this type of injury to the females, thus affecting overall fertility of *E. saccharina* reared on these diets. The reduced fertility of cold exposed *E. saccharina* females has important implications in terms of the SIT, because one of the major concerns in an SIT program is that laboratory-reared females may compete with wild females to mate with the F1 adults released [40]. If this did occur, the cold chilled females, even if fertile, would be only minimally competitive, thus decreasing the chances of them mating with the released sterile males.

## 5. Conclusions

When a lower concentration of the sterol mix (0.2 gC: 0.2 gS) was incorporated into the MS diet, the diet exhibited significant beneficial effects on the growth and development of *E. saccharina*. Feeding *E. saccharina* larvae on this diet reduced its developmental time (70% pupae produced by day 20) as compared with the control diet (15% pupae produced by day 20). Reduction in the development period will accelerate insect production in the SASRI insectary thus providing more insect generations per year that can be sterilised and released into the field.

The addition of cryoprotectants into the MS diet, particularly at a concentration mix of 0.2 gP: 0.2 gT, enhanced *E. saccharina* adult male cold hardiness in terms of chill coma recovery time. Feeding *E. saccharina* larvae on this diet reduced chill coma recovery time of *E. saccharina* male moths by 49.8 s when compared with that of the control diet, and by 101.0 s compared with the diet formulations containing L-proline. Production of high-quality males that recover quickly from cold stress and continue mating, will increase their mating competitiveness with wild males when released into the field.

The impact of compromised *E. saccharina* female fertility—as a result of chill coma treatment—on the mating success of released sterile males in an SIT program requires further investigation, as it could be beneficial in SIT programs by impairing mating competitiveness of mass-reared and chilled females, thus allowing the released sterilised males to mate with the more competitive wild females, thus impacting the wild population more.

Although the MS (0.2 gP:0.2 gT) diet exposed favourable rearing characteristic for *E. saccharina* larvae fed on it, the MS (0.2 gC:0.2 gS) is the preferred choice to replace the current diet used to rear *E. saccharina* at SASRI, as it reduced the larval growth period by 60% compared to the other diets in this study, without having adverse effects on key quality parameters of *E. saccharina*.

## Figures and Tables

**Figure 1 insects-13-00314-f001:**
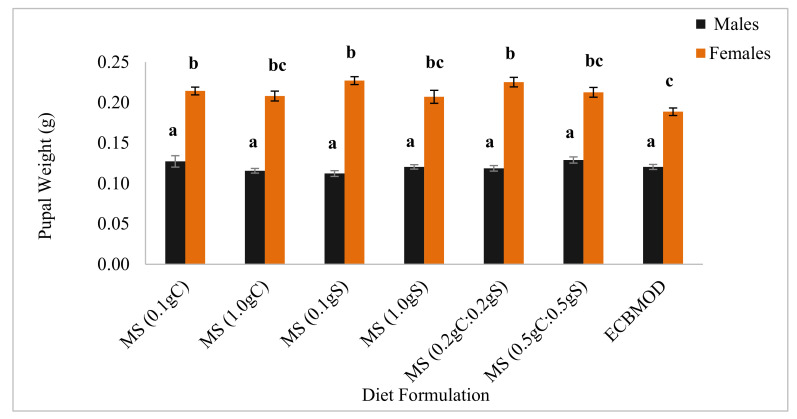
Mean (±SE) *Eldana saccharina* male and female pupal weights [*n* = 30] at harvest (Day 27) for the MS (0.1 gC), MS (1.0 gC), MS (0.1 gS), MS (1.0 gS), MS (0.2 gC:0.2 gS), MS (0.5 gC:0.5 gS) and ECBMOD control diets. Different lower-case letters above the graph histogram bars indicate that mean differences are significant at the 0.05 level.

**Figure 2 insects-13-00314-f002:**
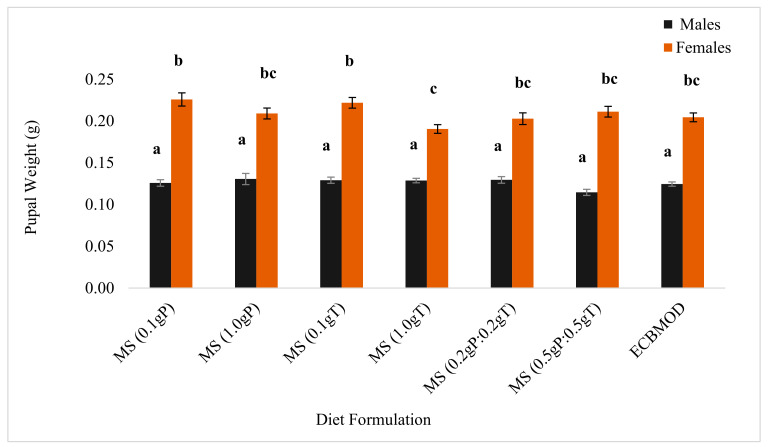
Mean (±SE) *Eldana saccharina* male and female pupal weights [*n* = 30] at harvest (Day 27) for the MS (0.1 gP), MS (1.0 gP), MS (0.1 gT), MS (1.0 gT), MS (0.2 gP:0.2 gT), MS (0.5 gP:0.5 gT) and ECBMOD control diets. Different lower-case letters above the graph histogram bars indicate that mean differences are significant at the 0.05 level.

**Figure 3 insects-13-00314-f003:**
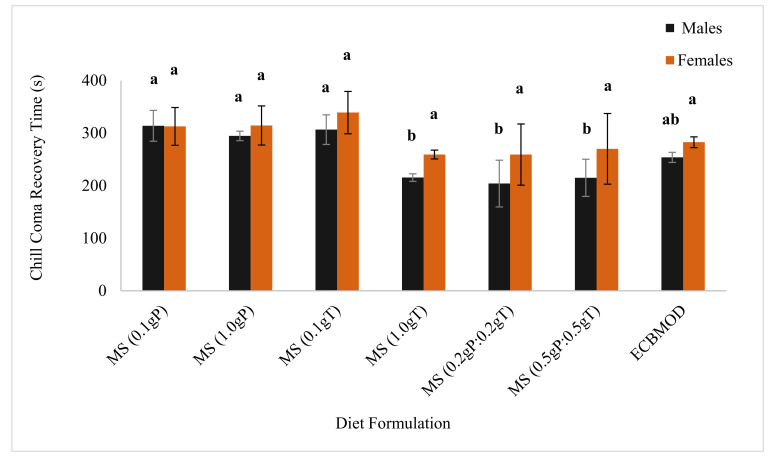
Mean (±SE) chill coma recovery time of *Eldana saccharina* male and female moths [*n* = 30] from the MS (0.1 gP), MS (1.0 gP), MS (0.1 gT), MS (1.0 gT), MS (0.2 gP:0.2 gT), MS (0.5 gP:0.5 gT) and ECBMOD control diets. Different lower-case letters above the graph histogram bars indicate that mean differences are significant at the 0.05 level.

**Table 1 insects-13-00314-t001:** *Eldana saccharina* MS diet with incorporation of different sterol components and ECBMOD control diet formulations.

MS Diets Containing Sterols vs. Control Diet (ECBMOD)								
	Diet Name	MS (0.1 gC)	MS (1.0 gC)	MS (0.1 gS)	MS (1.0 gS)	MS (0.2 gC:0.2 gS)	MS (0.5 gC:0.5 gS)	ECBMOD (Control)
Ingredients	Unit	Wt.	Wt.	Wt.	Wt.	Wt.	Wt.	Wt.
Carrageenan gel	g	15.00	15.00	15.00	15.00	15.00	15.00	
Agar powder	g							4.60
Lucerne meal	g	250.00	250.00	250.00	250.00	250.00	250.00	
Rabbit meal	g							226.40
Wheat bran	g	55.80	55.80	55.80	55.80	55.80	55.80	56.60
Yeast extract	g							3.40
Ground chickpea	g	53.00	53.00	53.00	53.00	53.00	53.00	56.60
Full cream milk powder	g	7.60	7.60	7.60	7.60	7.60	7.60	22.60
Whole egg powder	g	28.40	28.40	28.40	28.40	28.40	28.40	28.20
Sucrose	g	66.20	66.20	66.20	66.20	66.20	66.20	64.60
Sodium chloride	g							0.60
Nipagin	g	6.40	6.40	6.40	6.40	6.40	6.40	6.40
Sodium propionate	g	10.40	10.40	10.40	10.40	10.40	10.40	10.40
Oxytetracycline	g	2.00	2.00	2.00	2.00	2.00	2.00	2.00
Ascorbic acid	g	6.40	6.40	6.40	6.40	6.40	6.40	6.40
Acetic acid	mL	8.00	8.00	8.00	8.00	8.00	8.00	8.00
Citric acid	g	2.60	2.60	2.60	2.60	2.60	2.60	2.60
Tri-sodium citrate	g	2.60	2.60	2.60	2.60	2.60	2.60	2.60
Vit + min premix	g	0.80	0.80	0.80	0.80	0.80	0.80	
Cholesterol	g	0.10	1.00			0.20	0.50	
Stigmasterol	g			0.10	1.00	0.20	0.50	
Total		515.30	516.20	515.30	515.60	515.60	516.20	502.00
Water for agar	mL							500.00
Water balance	mL	1500.00	1500.00	1500.00	1500.00	1500.00	1500.00	1000.00
Total diet volume	mL	2015.30	2016.20	2015.30	2016.20	2015.60	2016.20	2002.00

**Table 2 insects-13-00314-t002:** *Eldana saccharina* MS diet with incorporation of different cryoprotectant components and ECBMOD control diet formulations.

MS Diets Containing Cryoprotectants vs. Control Diet (ECBMOD).								
	Diet Name	MS (0.1 gP)	MS (1.0 gP)	MS (0.1 gT)	MS (1.0 gT)	MS (0.2 gP:0.2 gT)	MS (0.5 gP:0.5 gT)	ECBMOD (Control)
Ingredients	Unit	Wt.	Wt.	Wt.	Wt.	Wt.	Wt.	Wt.
Carrageenan gel	g	15.00	15.00	15.00	15.00	15.00	15.00	
Agar powder	g							4.60
Lucerne meal	g	250.00	250.00	250.00	250.00	250.00	250.00	
Rabbit meal	g							226.40
Wheat bran	g	55.80	55.80	55.80	55.80	55.80	55.80	56.60
Yeast extract	g							3.40
Ground chickpea	g	53.00	53.00	53.00	53.00	53.00	53.00	56.60
Full cream milk powder	g	7.60	7.60	7.60	7.60	7.60	7.60	22.60
Whole egg powder	g	28.40	28.40	28.40	28.40	28.40	28.40	28.20
Sucrose	g	66.20	66.20	66.20	66.20	66.20	66.20	64.60
Sodium chloride	g							0.60
Nipagin	g	6.40	6.40	6.40	6.40	6.40	6.40	6.40
Sodium propionate	g	10.40	10.40	10.40	10.40	10.40	10.40	10.40
Oxytetracycline	g	2.00	2.00	2.00	2.00	2.00	2.00	2.00
Ascorbic acid	g	6.40	6.40	6.40	6.40	6.40	6.40	6.40
Acetic acid	mL	8.00	8.00	8.00	8.00	8.00	8.00	8.00
Citric acid	g	2.60	2.60	2.60	2.60	2.60	2.60	2.60
Tri-sodium citrate	g	2.60	2.60	2.60	2.60	2.60	2.60	2.60
Vit + min premix	g	0.80	0.80	0.80	0.80	0.80	0.80	
L-Proline	g	0.10	1.00			0.20	0.50	
Trehalose	g			0.10	1.00	0.20	0.50	
Total		515.30	516.20	515.30	515.60	515.60	516.20	502.00
Water for agar	mL							500.00
Water balance	mL	1500.00	1500.00	1500.00	1500.00	1500.00	1500.00	1000.00
Total diet volume	mL	2015.30	2016.20	2015.30	2016.20	2015.60	2016.20	2002.00

**Table 3 insects-13-00314-t003:** Mean (±SE) survival of *Eldana saccharina* life stages reared on the MS (0.1 gC), MS (1.0 gC), MS (0.1 gS), MS (1.0 gS), MS (0.2 gC:0.2 gS), MS (0.5 gC:0.5 gS) and ECBMOD control diets from inoculation of neonates to first pupation [*n* = 15] at day 20 and full pupal production at harvest [*n* = 85] (Day 27).

Diet Formulation	% Survival at First Pupation (Day 20)	% Survival at Full Pupal Development (Day 27)
MS (0.1 gC)	100.00 ± 0	98.00 ± 1.01
MS (1.0 gC)	100.00 ± 0	100.00 ± 0
MS (0.1 gS)	93.00 ± 4.54	97.00 ± 1.28
MS (1.0 gS)	100.00 ± 0	96.00 ± 1.40
MS (0.2 gC:0.2 gS)	100.00 ± 0	99.00 ± 0.59
MS (0.5 gC:0.5 gS)	93.00 ± 4.54	99.00 ± 0.59
ECBMOD (Control)	100.00 ± 0	100.00 ± 0

**Table 4 insects-13-00314-t004:** Mean (±SE) distribution of *Eldana saccharina* life stages (percentage of the 3rd/4th instar larvae, 5th/6th instar larvae, pre-pupae, pupae and mortality) [*n* = 30] surviving on the MS (0.1 gC), MS (1.0 gC), MS (0.1 gS), MS (1.0 gS), MS (0.2 gC:0.2 gS), MS (0.5 gC:0.5 gS) and ECBMOD control diets, 20 days after inoculation. Means within columns with different lower-case letters indicate significant differences (*p* < 0.05).

	Life Stage Distribution (%)					
Diet Formulation	1st/2nd Instar *	3rd/4th Instar	5th/6th Instar	Pre-Pupae	Pupae	Mortality
MS (0.1 gC)	0 ± 0	0 ± 0 b	27.0 ± 0.6 d	9.0 ± 0.6 b	64.0 ± 0.6 b	0 ± 0 c
MS (1.0 gC)	0 ± 0	0 ± 0 b	30.0 ± 0.6 c	7.0 ± 0.6 bc	63.0 ± 0.6 b	0 ± 0 c
MS (0.1 gS)	0 ± 0	0 ± 0 b	29.0 ± 0.6 cd	2.0 ± 0.6 d	62.0 ± 0.6 b	7.0 ± 0.6 a
MS (1.0 gS)	0 ± 0	0 ± 0 b	16.0 ± 0.6 f	12.0 ± 0.6 a	72.0 ± 0.6 a	0 ± 0 c
MS (0.2 gC:0.2 gS)	0 ± 0	0 ± 0 b	24.0 ± 0.6 e	6.0 ± 0.6 c	70.0 ± 0.6 a	0 ± 0 c
MS (0.5 gC:0.5 gS)	0 ± 0	0 ± 0 b	33.0 ± 0.6 b	7.0 ± 0.6 bc	55.0 ± 0.6 c	5.0 ± 0.6 b
ECBMOD (Control)	0 ± 0	35.0 ± 0.6 a	50.0 ± 0.6 a	0 ± 0.0 d	15.0 ± 0.6 d	0 ± 0 c

* No available *p*-value.

**Table 5 insects-13-00314-t005:** Mean (±SE) distribution of *Eldana saccharina* life stages (percentage of larvae, pre-pupae, pupae, moths and total mortality) [*n* = 170] recorded at the time of full pupal harvest (Day 27) on the MS (0.1 gC), MS (1.0 gC), MS (0.1 gS), MS (1.0 gS), MS (0.2 gC:0.2 gS), MS (0.5 gC:0.5 gS) and ECBMOD control diets. Means within columns with different lower-case letters indicate significant differences (*p* < 0.05).

	Life Stage Distribution (%)				
Diet Formulation	Larvae	Pre-Pupae	Pupae	Moths	Mortality
MS (0.1 gC)	2.0 ± 0.6 c	12.0 ± 0.6 b	82.0 ± 0.6 c	4.0 ± 0.6 a	0 ± 0 a
MS (1.0 gC)	0 ± 0 c	0 ± 0 d	96.0 ± 0.6 a	4.0 ± 0.6 a	0 ± 0 a
MS (0.1 gS)	2.0 ± 0.6 c	11.0 ± 0.6 bc	86.0 ± 0.6 b	1.0 ± 0.6 b	0 ± 0 a
MS (1.0 gS)	6.0 ± 0.6 b	9.0 ± 0.6 c	82.0 ± 0.6 c	2.0 ± 0.6 ab	1.0 ± 0.6 a
MS (0.2 gC:0.2 gS)	6.0 ± 0.6 b	11.0 ± 0.6 bc	82.0 ± 0.6 c	0 ± 0 b	1.0 ± 0.6 a
MS (0.5 gC:0.5 gS)	7.0 ± 0.6 b	19.0 ± 0.6 a	73.0 ± 0.6 d	0 ± 0 b	1.0 ± 0.6 a
ECBMOD (Control)	19.0 ± 0.6 a	9.0 ± 0.6 c	70.0 ± 0.6 e	0 ± 0 b	2.0 ± 0.6 a

**Table 6 insects-13-00314-t006:** Mean (±SE) emergence and sex ratio (Male: Female) of *Eldana saccharina* moths [*n* = 130] from pupae harvested from the MS (0.1 gC), MS (1.0 gC), MS (0.1 gS), MS (1.0 gS), MS (0.2 gC:0.2 gS), MS (0.5 gC:0.5 gS) and ECBMOD control diets. Means within columns with different lower-case letters indicate significant differences (*p* < 0.05).

Diet Formulation	Moth Emergence (%)	Sex Ratio (M:F)
MS (0.1 gC)	96.0 ± 0.6 b	1.3 ± 0.1 a
MS (1.0 gC)	96.0 ± 0.6 b	1.1 ± 0.1 a
MS (0.1 gS)	100.0 ± 0.3 a	0.8 ± 0.1 b
MS (1.0 gS)	94.0 ± 0.6 c	0.9 ± 0.1 b
MS (0.2 gC:0.2 gS)	100.0 ± 0.3 a	1.2 ± 0.1 a
MS (0.5 gC:0.5 gS)	92.0 ± 0.6 c	1.1 ± 0.1 a
ECBMOD (Control)	97.0 ± 0.6 b	0.8 ± 0.1 b

**Table 7 insects-13-00314-t007:** Mean (±SE) emergence and sex ratio (Male: Female) of *Eldana saccharina* moths [*n* = 130] from pupae harvested from the MS (0.1 gP), MS (1.0 gP), MS (0.1 gT), MS (1.0 gT), MS (0.2 gP:0.2 gT), MS (0.5 gP:0.5 gT) and ECBMOD control diets. Means within columns with different lower-case letters indicate significant differences (*p* < 0.05).

Diet Formulation	Moth Emergence (%)	Sex Ratio (M:F)
MS (0.1 gP)	95.0 ± 0.6 b	0.9 ± 0.1 c
MS (1.0 gP)	96.0 ± 0.6 b	1.3 ± 0.1 b
MS (0.1 gT)	89.0 ± 0.6 c	1.3 ± 0.1 b
MS (1.0 gT)	100.0 ± 0.3 a	1.2 ± 0.1 b
MS (0.2 gP:0.2 gT)	91.0 ± 0.6 c	1.0 ± 0.1 bc
MS (0.5 gP:0.5 gT)	96.0 ± 0.6 b	1.0 ± 0.1 bc
ECBMOD (Control)	89.0 ± 0.6 c	1.7 ± 0.1 a

**Table 8 insects-13-00314-t008:** Mean (±SE) mating frequency of chill coma exposed and non-chill coma exposed *Eldana saccharina* male and female moths [*n* = 15], after being paired with moths from the same formulations but not exposed to chill coma, from the MS (0.1 gP), MS (1.0 gP), MS (0.1 gT), MS (1.0 gT), MS (0.2 gP:0.2 gT), MS (0.5 gP:0.5 gT) and ECBMOD control diets.

	Male Mating Frequency		Female Mating Frequency	
Diet Formulation	Chill Coma Exposed	Non-Chill Exposed	Chill Coma Exposed	Non-Chill Exposed
MS (0.1 gP)	3.00 ± 1.00	3.00 ± 0.67	3.00 ± 0.58	2.00 ± 0.33
MS (1.0 gP)	3.00 ± 0.33	3.00 ± 0.33	3.00 ± 0.33	3.00 ± 0.33
MS (0.1 gT)	3.00 ± 0.67	4.00 ± 0.33	3.00 ± 0.33	2.00 ± 0.33
MS (1.0 gT)	4.00 ± 0.33	4.00 ± 0.58	2.00 ± 0.33	2.00 ± 0.33
MS (0.2 gP:0.2 gT)	3.00 ± 0.67	4.00 ± 0.33	2.00 ± 0.68	2.00 ± 0.33
MS (0.5 gP:0.5 gT)	3.00 ± 0.58	3.00 ± 0.33	2.00 ± 0.6	2.00 ± 0.33
ECBMOD (Control)	3.00 ± 0.33	4.00 ± 0.33	2.00 ± 0.33	2.00 ± 0.33

**Table 9 insects-13-00314-t009:** Mean (±SE) fecundity of *Eldana saccharina* female moths [*n* = 15] and their egg fertility after mating with moths from the same formulations, from the MS (0.1 gC), MS (1.0 gC), MS (0.1 gS), MS (1.0 gS), MS (0.2 gC:0.2 gS), MS (0.5 gC:0.5 gS) and ECBMOD control diets.

Diet Formulation	Fecundity (*n*)	Fertility (%)
MS (0.1 gC)	879.00 ± 40.32	94.00 ± 5.65
MS (1.0 gC)	955.00 ± 28.81	96.00 ± 1.57
MS (0.1 gS)	827.00 ± 90.30	98.00 ± 2.03
MS (1.0 gS)	922.00 ± 105.06	98.00 ± 1.34
MS (0.2 gC:0.2 gS)	891.00 ± 148.86	91.00 ± 7.59
MS (0.5 gC:0.5 gS)	906.00 ± 20.25	100.00 ± 0.15
ECBMOD (Control)	789.00 ± 48.66	94.00 ± 1.77

**Table 10 insects-13-00314-t010:** Mean (±SE) fecundity of chill coma exposed and non-chill exposed *Eldana saccharina* female moths [*n* = 15] and their egg fertility after mating with males from the same formulations but not exposed to chill coma, from the MS (0.1 gP), MS (1.0 gP), MS (0.1 gT), MS (1.0 gT), MS (0.2 gP:0.2 gT), MS (0.5 gP:0.5 gT) and ECBMOD control diets. Means followed by different lower-case letters in a line indicate significant differences (*p* < 0.05).

	Female Fecundity (*n*)		Female Fertility (%)	
Diet Formulation	Chill Coma Exposed	Non-Chill Exposed	Chill Coma Exposed	Non-Chill Exposed
MS (0.1 gP)	710.00 ± 7.22	936.00 ± 39.50	10.00 ± 9.68 b	90.00 ± 7.49 a
MS (1.0 gP)	776.00 ± 240.01	733.00 ± 82.50	30.00 ± 2.00 a	95.00 ± 4.85 a
MS (0.1 gT)	765.00 ± 54.58	800.00 ± 46.00	42.00 ± 6.28 a	78.00 ± 21.9 a
MS (1.0 gT)	783.00 ± 47.61	786.00 ± 49.19	41.00 ± 5.99 a	80.00 ± 7.73 a
MS (0.2 gP:0.2 gT)	813.00 ± 150.82	853.00 ± 85.50	34.00 ± 5.82 a	95.00 ± 1.31 a
MS (0.5 gP:0.5 gT)	723.00 ± 30.84	732.00 ± 60.50	14.00 ± 7.59 b	64.00 ± 13.19 a
ECBMOD (Control)	815.00 ± 25.42	841.00 ± 41.15	31.00 ± 3.72 a	88.00 ± 6.45 a

## Data Availability

Data supporting the reported results are the property of SASRI, and can be requested from the Programme Manager listed under the Crop Protection Program of SASRI (https://sasri.org.za, accessed on 16 February 2022).

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
