# Peer review of "The Addition of Sterols and Cryoprotectants to Optimize a Diet Developed for Eldana saccharina Walker (Lepidoptera: Pyralidae) Using the Carcass Milling Technique"

_insects, 2022, doi:10.3390/insects13040314_

Round 1

Reviewer 1 Report

Good work,

This is a valuable research work.

Materials and methods section.

Please add product/part numbers to materials used.

Thank you

Reviewer 2 Report

This paper brings valuable data on diet optimization to improve effectiveness of the mass rearing program for Eldana saccharina moths. The paper is very well written and contains detailed information on insect handling in every step of the rearing process and in every experiment. In fact, the paper is written as technical report that focuses on reliability and replicability of presented results (potentially applicable in mass rearing program) rather than on scientific novelty. The paper is perfectly suitable for Insects and I think that it can be published almost as is.

Below, I have written a list of minor comments for the authors to consider.   

L19: 90-101 seconds. Written like this, the reader can hardly appreciate the magnitude of this change. Consider saying; shortened from X to Y, or shortened X-fold, or shortened by X %.

L19-21: I am not sure whether this is the only possible interpretation of the results. First, it is obvious that chill coma drastically reduces the fertility irrespective of the diet. When considering the ECBMOD diet as standard (without additional sterols or cryoprotectants), the reduction goes to 31% (Table 10). Second, some diet compositions (i.e. 0.1T, 1.0T) resulted in higher fertility than standard (41 and 42%, resp.). Third, other diet compositions had clearly negative effect – reducing the fertility much below standard (perhaps because you combine two stresses: chill coma and toxic compound in the diet).

Further to this point:

The chill coma (0°C for 2 h) was relatively severe. Probably unlikely to occur in nature where the moth lives. But I agree that such temperatures might be encountered in mass reared insects at storage or shipment.

The toxic effects of (high doses) of cryoprotectants added to insect diet are described in literature. It is interesting to see again in this paper, that DESPITE this toxicity, the cold tolerance improves (here shown as reduction of the CCRT).

Table 1: Looking at the composition of diets one can see that ALL diets contain a number of components that likely have some sterols. This is not a big problem because you can still observe the effects of added sterols. Have you any idea on basal sterol contents in the diets? And, the same question for Table 2, cryoprotectants.

Table 10: The value 14.00 ± 7.59a for female fertility in chill coma exposed insects fed on MS (0.5gP:0.5gT) diet has a flanking letter 'a'. Is this correct?

L603: perhaps say: enhanced the cold tolerance (i.e. reduced the CCRT)
